# Identification of New Copy Number Variation and the Evaluation of a CNV Detection Tool for NGS Panel Data in Polish Familial Hypercholesterolemia Patients

**DOI:** 10.3390/genes13081424

**Published:** 2022-08-10

**Authors:** Lena Rutkowska, Iwona Pinkier, Kinga Sałacińska, Łukasz Kępczyński, Dominik Salachna, Joanna Lewek, Maciej Banach, Paweł Matusik, Ewa Starostecka, Andrzej Lewiński, Rafał Płoski, Piotr Stawiński, Agnieszka Gach

**Affiliations:** 1Department of Genetics, Polish Mother’s Memorial Hospital—Research Institute, 93-338 Lodz, Poland; 2Department of Preventive Cardiology and Lipidology, Medical University of Lodz, 90-419 Lodz, Poland; 3Department of Cardiology and Congenital Diseases of Adults, Polish Mother’s Memorial Hospital—Research Institute, 93-338 Lodz, Poland; 4Cardiovascular Research Centre, University of Zielona Gora, 65-417 Zielona Gora, Poland; 5Department of Pediatrics, Pediatric Obesity and Metabolic Bone Diseases, Faculty of Medical Sciences in Katowice, Medical University of Silesia, 40-055 Katowice, Poland; 6Department of Endocrinology and Metabolic Diseases, Polish Mother’s Memorial Hospital—Research Institute, 93-338 Lodz, Poland; 7Department of Endocrinology and Metabolic Diseases, Medical University of Lodz, 90-419 Lodz, Poland; 8Department of Medical Genetics, Medical University of Warsaw, 02-106 Warsaw, Poland

**Keywords:** inherited disorder, familial hypercholesterolemia, copy number variation (CNV), *LDLR* gene, genetic basis, bioinformatic tool, DECoN, phenotype-genotype correlation, panel next generation sequencing (NGS), multiplex ligation-dependent amplification (MLPA)

## Abstract

Familial hypercholesterolemia (FH) is an inherited, autosomal dominant metabolic disorder mostly associated with disease-causing variant in *LDLR*, *APOB* or *PCSK9*. Although the dominant changes are small-scale missense, frameshift and splicing variants, approximately 10% of molecularly defined FH cases are due to copy number variations (CNVs). The first-line strategy is to identify possible pathogenic SNVs (single nucleotide variants) using multiple PCR, Sanger sequencing, or with more comprehensive approaches, such as NGS (next-generation sequencing), WES (whole-exome sequencing) or WGS (whole-genome sequencing). The gold standard for CNV detection in genetic diagnostics are MLPA (multiplex ligation-dependent amplification) or aCGH (array-based comparative genome hybridization). However, faster and simpler analyses are needed. Therefore, it has been proposed that NGS data can be searched to analyze CNV variants. The aim of the study was to identify novel CNV changes in FH patients without detected pathogenic SNVs using targeted sequencing and evaluation of CNV calling tool (DECoN) working on gene panel NGS data; the study also assesses its suitability as a screening step in genetic diagnostics. A group of 136 adult and child patients were recruited for the present study. The inclusion criteria comprised at least “possible FH” according to the Simon Broome diagnostic criteria in children and the DLCN (Dutch Lipid Clinical Network) criteria in adults. NGS analysis revealed potentially pathogenic SNVs in 57 patients. Thirty selected patients without a positive finding from NGS were subjected to MLPA analysis; ten of these revealed possibly pathogenic CNVs. Nine patients were found to harbor exons 4–8 duplication, two harbored exons 6–8 deletion and one demonstrated exon 9–10 deletion in *LDLR*. To test the DECoN program, the whole study group was referred for bioinformatic analysis. The DECoN program detected duplication of exons 4–8 in the *LDLR* gene in two patients, whose genetic analysis was stopped after the NGS step. The integration of the two methods proved to be particularly valuable in a five-year-old girl presenting with extreme hypercholesterolemia, with both a pathogenic missense variant (c.1747C>T) and exons 9–10 deletion in *LDLR*. This is the first report of a heterozygous deletion of exons 9 and 10 co-occurring with SNV. Our results suggest that the NGS-based approach has the potential to identify large-scale variation in the *LDLR* gene and could be further applied to extend CNV screening in other FH-related genes. Nevertheless, the outcomes from the bioinformatic approach still need to be confirmed by MLPA; hence, the latter remains the reference method for assessing CNV in FH patients.

## 1. Introduction

Familial hypercholesterolemia (FH) is an inherited, autosomal dominant metabolic disorder mostly caused by disease-causing variant in *LDLR* (low-density lipoprotein receptor gene), *APOB* (apolipoprotein B gene) or *PCSK9* (proprotein convertase subtilisin/kexin type 9 gene). The respective prevalence of heterozygous FH (HeFH) and homozygous FH (HoFH) was initially thought to be 1:500 and 1:1,000,000, but these figures have been revised following the rapid development of genetics and increased awareness of the disease. Nowadays, it has been estimated that HeFH affects 1:313 individuals worldwide, and HoFH is still being ultrarare condition with prevalence of 1:160–400,000 [1,2]. The clinical hallmarks of FH are high total and LDL cholesterol levels; these are directly associated with increased cardiovascular risk, with the main clinical manifestation being ischemic heart disease (IHD). Early detection and initiation of lipid-lowering treatment is crucial for ASCVD (Atherosclerotic Cardiovascular Disease) prevention, with an objective of LDL cholesterol ˂55 (1.42 mmol/L) or 70 mg/dL (1.81 mmol/L) depending on the risk and a decrease of at least 50% [3,4].

The most widely adopted diagnostic algorithms are Dutch Lipid Clinical Network Criteria (DLCN), Simon Broome (SB) and Making Early Diagnosis Prevents Early Death (MEDPED). However, regardless of the criteria, genetic testing is unequivocal, and forms a central part of any diagnosis [5]. 

The FH phenotype is caused by loss of function variants in the *LDLR* gene in 60–80% patients, followed by those in *APOB* (5–10%) or by gain-of-function variants in *PCSK9* (<1%) [6]. Rarely, the patient harbors an ultra-rare variant in the *APOE* gene or one of a wide group of candidate genes (*LDLRAP1, LIPA, SCAP, GPIHBP1* or *STAP1*). The dominant changes are small-in-scale missense, frameshift and splicing variants but approximately 10% of molecular defined FH are due to copy number variations (CNVs). CNVs are genomic structural variants that include deletions and duplications larger than 50 bp in size [7]. CNV regions are found ubiquitously throughout the human genome, and encompass about 4.8–9.7% of its total sequence [8]. Depending on the size and genomic localization, CNVs can have a range of functional consequences ranging from neutral to adaptive to maladaptive traits [9]. In the last decade, maladaptive CNVs have been found to play a role in many human diseases such as autism, schizophrenia or Crohn’s disease. Their possible role in dyslipidemias is still being investigated; research to date has focused primarily on causal changes for FH, but this may broaden as techniques evolve. A 2018 review found about 56 unique deletions and 27 unique duplications had been detected in the *LDLR* gene [9]. Such a high number is probably related to the fact that there are 98 *Alu* repeats within the gene [10]; it was also found that *Alu* repeats represent 65% of *LDLR* intronic sequences, and 85% of genomic sequence outside exon–intron junctions [11]. *LDLR* is hence especially susceptible to CNV rearrangements with breakpoints mostly located within the introns, leading to whole-exon events [10].

The development of new, cost-effective, rapid and efficient molecular techniques is now extremely important. The first-line strategy is to identify possibly pathogenic SNVs (single nucleotide variants) using multiple PCR, Sanger sequencing or more comprehensive approaches, such as panel next-generation sequencing (NGS), whole-exome sequencing (WES) or whole-genome sequencing (WGS). Conventional Sanger sequencing significantly limits the scope of the study, while techniques such as WES or WGS provide enormous amounts of data which are often challenging to interpret. Hence, the optimal technique within the reach of most research laboratories is targeted NGS. NGS technology allows the detection of single-nucleotide variants and small deletion/insertion variants for Mendelian conditions. However, when diagnosing FH, it should be noted that large CNV variants make up about 10% of the genetic background, and their detection is a necessary step in a comprehensive genetic diagnostics strategy. 

The gold standard for CNV detection in genetic diagnostics are multiplex ligation-dependent amplification (MLPA) or array-based comparative genome hybridization (aCGH) [12]. However, to provide simpler and quicker analysis, new approaches based on using NGS data to analyze CNV variants have been proposed. A number of bioinformatic tools have been developed to analyze post-NGS data, but not all of them show adequate sensitivity or specificity, or acceptable false discovery rates. The detection of large rearrangements from targeted NGS data is still complicated by issues intrinsic to the technology, such as short read lengths [12].

The aim of the study was to search novel CNV changes in FH patients without detected pathogenic SNVs in targeted sequencing and evaluation of CNV calling tool (DECoN) working on NGS data; it also assesses the suitability of the method as a screening step in genetic diagnostics.

## 2. Materials and Methods

### 2.1. Patients

A total of 136 adult and children patients from the EAS-FHSC Regional Center for Rare Diseases at the Polish Mother’s Memorial Hospital—Research Institute (PMMHRI) in Poland were recruited for the present study [13]. The inclusion criteria were the status of at least “possible FH” according to the Simon Broome diagnostic criteria in children and the Dutch Lipid Clinical Network Criteria in adults. The study was conducted in accordance with the Declaration of Helsinki and approved by the PMMHRI Ethics Committee (opinion number 15/2016, date of approval 12 January 2016). Informed consent was obtained from all subjects involved in the study.

### 2.2. NGS Analysis

Genomic DNA was isolated from peripheral blood samples using a MagCore automatic nucleic acid extractor (RBC Bioscience, New Taipei City, Taiwan). The entire procedure of preparing the libraries for NGS sequencing was conducted in accordance with the manufacturer’s protocol and was described in detail in the cited resource [14]. A custom NGS panel containing 21 causative and candidate genes linked to familial hypercholesterolemia and other primary dyslipidemias (*ABCA1, ABCG5, ABCG8, APOA5, APOB, APOC2, APOE, CYP7A1, GPIHBP1, LCAT, LDLR, LDLRAP1, LIPA, LMF1, LMNA, LPL, PCSK9, PPARG, SCAP, SREBF2, STAP1*). The obtained NGS data were processed and analyzed by VariantStudio Software. The pathogenicity of the variants was determined in silico using web-based software, such as PolyPhen2, SIFT and Mutation Taster. Searches for phenotype–genotype correlations were evaluated using PubMed, LOVD or VARSOME databases. Variants were classified according to current American College of Medical Genetics and Genomics (ACMG) guidelines [15]. The presence of selected variants was confirmed by bidirectional Sanger sequencing on a 3500 Series Genetic Analyzer (Applied Biosystems, Waltham, MA, USA). DNA Variant Analysis was performed using Mutation Surveyor V5.1.0 software (SoftGenetics, State College, PA, USA). 

### 2.3. MLPA Analysis

The MLPA analysis was conducted in accordance with the manufacturer’s protocol (MRC-Holland, Amsterdam, The Netherlands) using the SALSA MLPA Probemix P062 *LDLR*. The PCR products were combined with labelled size standard (GeneScan 500 LIZ Size Standard; Applied Biosystems, Waltham, MA, USA) and separated by capillary electrophoresis on a 3500 Series Genetic Analyzer (Applied Biosystems, Waltham, MA, USA). The P062 kit contains 20 probes for *LDLR*, one flanking probe for upstream of *LDLR* and 12 reference probes for gene loci on alternative autosomal chromosomes. The GeneMarker v1.95 (SoftGenetics, State College, PA, USA) was used to perform pattern comparison of peak height between patient samples and control samples. Each amplification yields a pattern composed of fluorescent FAM-labeled peaks, with each peak corresponding to a specific genomic DNA locus. 

### 2.4. CNV Calling

CNV screening of NGS data was performed using the DECoN bioinformatic tool. To generate a coverage metrics of exons the bam files and bed files obtained from targeted NGS were uploaded. Then, quality checks were performed to flag any samples or exons where exon CNV calling may be suboptimal. Both exons and samples were evaluated based on their median coverage level and their mutual correlation. Samples without a high correlation with others in the set are likely to have suboptimal detection across the entire target. The default value was 0.98. After rejecting non-compliant samples, the exon CNV calling was performed. All generated calls were collected in a summary table containing CNV ID, sample ID, correlation score (the maximum correlation between the test sample and any other sample in the full set of bam files), N.comp (the number of samples used as the reference set), Start.b (the number of the first exon in the call from the analyzed bed file), End.b (the number of the last exon in the call from the analyzed bed file), CNV type, N.exons (number of exons encompassed by the call), Start (the start position of the call from the analyzed bed file), End (the end position of the call from the analyzed bed file), Chromosome, Genomic ID, the Bayes Factor, Reads.expected, Reads.observed, Reads.ratio and name of the affected gene. The bioinformatics content was carried out in cooperation with research team from Medical University of Warsaw, Poland.

## 3. Results

A group of 136 pediatric and adult patients with a clinical suspicion of familial hypercholesterolemia (FH) was recruited for the present study. Potentially pathogenic SNVs were identified in 57 patients, who were referred to Sanger sequencing confirmation. Thirty selected patients without a positive NGS finding were referred for MLPA analysis; the selective referral to MLPA analysis was dictated by economic considerations and only patients with highest LDL cholesterol were selected. The MLPA analysis revealed possibly pathogenic CNVs in 10 patients. To test the DECoN program, the whole study group was referred for bioinformatic analysis. DECoN detected duplication of exons 4–8 in *LDLR* gene in two patients whose genetic analysis was stopped after the NGS step. Both cases were validated with MLPA. The research process is presented in Figure 1.

Among twelve CNV-positive patients, six were children and six adults. Patient age, sex, lipid profile and obtained genetic results are shown in Table 1.

Nine patients demonstrated exons 4–8 heterozygous duplication, two showed exons 6–8 heterozygous deletion and one patient exons 9–10 heterozygous deletion. Examples of the MLPA images are presented in Figure 2.

### 3.1. Duplication of Exons 4–8 of LDLR

Patients 1 and 2 represent a family case (son and mother); the 15-year-old boy (weight: 44.5 kg; high: 156 cm) was admitted to hospital with a TC level of 293 mg/dL (3.31 mmol/L) and LDL-c of 253 mg/dL (6.54 mmol/L). Implemented statin therapy resulted in a reduction in the TC level to 224 mg/dL (5.79 mmol/L) and LDL-c to 165 mg/dL (4.27 mmol/L). Lipid testing of close family members reveals a history of hypercholesterolemia in the mother [TC 287 mg/dL (3.24 mmol/L) and LDL-c 203 mg/dL (5.25 mmol/L)] and siblings (18-year-old brother and 23-year-old sister). A physical examination revealed no abnormalities.

Patients 3 and 4 are mother and daughter; a 3-year-old girl demonstrated hypercholesterolemia with a TC level of 277 mg/dL (7.16 mmol/L) and LDL-c of 222 mg/dL (5.74 mmol/L). NGS analysis revealed the presence of a heterozygous missense variant c.56C > G (rs3135506) in *APOA5* gene. This variant has previously been associated with an increased risk of hypertriglyceridemia [16,17], but current reports are more likely to indicate this as functional polymorphism [18,19]. As our patient did not demonstrate high TG concentration and the c.56C > G variant still did not explain the cause of hypercholesterolemia, further genetic diagnostics was required.

Case number 5 is a 64-year-old woman suffering from hypertension, generalized atherosclerosis and dizziness. Her lipid profile presented combined hyperlipidemia with a TC level of 424 mg/dL (10.96 mmol/L), LDL-c of 344 mg/dL (8.90 mmol/L) and TG of 344 mg/dL (3.88 mmol/L). At that time, the patient began pharmacological lipid-lowering treatment with rosuvastatin and ezetimibe (each 10 mg daily doses). After more than a year of treatment, the cholesterol parameters dropped to 271 mg/dL (7.01 mmol/L) for TC and 145 mg/dL (3.75 mmol/L) for LDL-c. The patient’s daughter, with LDL-c level over 190 mg/dL, was referred for genetic testing.

Patient 8, a 16-year-old girl, presented severe hypercholesterolemia with a TC level of 440 mg/dL (11.38 mmol/L) and LDL-c of 267 mg/dL (6.90 mmol/L). The girl was started on atorvastatin. The patient’s mother was also symptomatic with TC 290 mg/dL (7.50 mmol/L) and LDL-c 288 mg/dL (7.45 mmol/L), during statin therapy.

Patient 10 is 68-year-old women with chronic coronary syndrome after coronary artery bypass graft surgery. Her family history indicates the presence of severe lipid abnormalities- two brothers experienced sudden heart attacks at the ages of 25 and 39. The patient was treated with fenofibrate; lipid parameters during the drug treatment: TC 196 mg/dL (5.07 mmol/L), LDL-c 140 mg/dL (3.62 mmol/L), HDL 37 mg/dL (0.96 mmol/L), TG 104 mg/dL (1.17 mmol/L). 

Case number 11 is a 28-year-old boy with severe hypercholesterolemia identified at age 8, with a TC level of 329 mg/dL (8.51 mmol/L), LDL-c of 270 mg/dL (6.98 mmol/L), HDL of 50 mg/dL (1.29 mmol/L) and TG of 43 mg/dL (0.49 mmol/L), at that time. Since the age of 18, treatment with atorvastatin, rosuvastatin, simvastatin and ezetimibe has been implemented. Due to muscle pains in the lower limbs and increased creatinine kinase level (452 IU/l), the hypolipidemic treatment was discontinued. A year later, the patient was referred for genetic testing, and his lipid parameters again showed severe lipid abnormalities [TC 333 mg/dL (8.61 mmol/L), LDL-c 290 mg/dL (7.50 mmol/L), HDL 39 mg/dL (1.01 mmol/L), TG 93 mg/dL (1.05 mmol/L)]. Currently, the patient reports a healthy lifestyle, diet, physical activity, and refuses drug treatment. The patient’s mother is also affected with a LDL-c level of 307 mg/dl (7.94 mmol/L).

Patient 12 is an untreated 10-year-old girl with a TC level of 371 mg/dL (9.59 mmol/L), LDL-c 274 mg/dL (7.09 mmol/L), HDL 83 mg/dL (2.15 mmol/L) and TG 70 mg/dL (0.79 mmol/L). The lipid parameters of the parents do not indicate inheritance of duplication [Father: TC 194 mg/dL (5.02 mmol/L), LDL-c 110 mg/dL (2.84 mmol/L), HDL 67 mg/dL (1.73 mmol/L), TG 87 mg/dL (0.98 mmol/L); Mother: TC 147 mg/dL (3.80 mmol/L), LDL-c 87 mg/dL (2.25 mmol/L), HDL 37 mg/dL (0.96 mmol/L), TG 173 mg/dL (1.95 mmol/L)]. No abnormalities were noted on physical examination.

### 3.2. Deletion of Exons 9–10 of LDLR

Case number 9 is a 5-year-old girl of normal weight (19 kg, 116 cm) showing extreme hypercholesterolemia with TC level of 745 mg/dL (19.27 mmol/L) and LDL-c of 693 mg/dL (17.92 mmol/L). The patient manifested characteristic phenotypic feature of hypercholesterolemia as xanthomas found on eyelid, Achilles tendon and knee area (Figure 3). The patient received treatment with 10 mg rosuvastatin, which resulted in a significant reduction in lipid parameters [TC 243 mg/dl (6.28 mmol/L), LDL-c 210 mg/dl (5.43 mmol/L]. In addition to extensive exon 9–10 deletion, NGS analysis revealed the presence of heterozygous missense variant c.1747C>T, p.(His583Tyr) in exon 12 of *LDLR* gene. The detected variant is reported in ClinVar (RCV000771547.7) and LOVD database (#0000093188) as likely pathogenic, corresponding to familial hypercholesterolemia. The performed MLPA analysis excluded the presence of exon 9–10 deletion in the girl’s parents, suggesting de novo origin.

### 3.3. Deletion of Exons 6–8 of LDLR

Patients 6 and 7 are mother and daughter; the 14-year-old girl was admitted to hospital with a TC level of 336 mg/dL (8.69 mmol/L), LDL-c 263 mg/dL (6.80 mmol/L), HDL 52 mg/dL (1.34 mmol/L) and TG 106 mg/dL (1.20 mmol/L). Her 35-year-old mother also has an abnormal lipid profile, with TC level of 368 mg/dL (9.52 mmol/L) and LDL-c 282 mg/dL (7.29 mmol/L); simvastatin treatment was implemented. Neither presented xanthomas or corneal arcus.

### 3.4. Bioinformatic Analysis

To confirm the changes detected in the MLPA and to search for new CNVs we processed post-NGS data by DECoN. The computational pipeline utilizes coverage depth of the captured regions and calculates a copy number ratio for each region. Based on variations in mean coverages between samples, the program identified two additional patients with suspected exon 4–8 duplication in the *LDLR* gene (Table 2). The presence of changes was confirmed by MLPA.

## 4. Discussion

Changes in copy number variation constitute the genetic background of many human diseases. It is known that familial hypercholesterolemia (FH) is mainly caused by point changes in *LDLR, APOB, PCKS9* or of a wide range of candidate genes. However, it should not be forgotten that about 10% of causal variation constitutes *LDLR* CNVs, so there is a high need to detect large-scale CNVs in addition to single nucleotide variants [9]. 

The first stage of the analysis examined all patients using an NGS custom panel, which identified disease-causing variants in 57 patients. A second-stage MLPA analysis, performed in 30 selected symptomatic patients, confirmed *LDLR* changes in 10 patients. Finally, the DECoN analysis detected CNV changes in two additional patients who had not been previously checked by the MLPA technique. Hence, 42% of the study group were found to have pathogenic SNV and 8.8% with causal CNV. The lowered percentage of CNV occurrence relative to the commonly reported 10%, can be considered as a study limitation. The most frequent alteration was heterozygous duplication of exons 4–8, found in nine individuals with an average concentration of TC and LDL-c of 348 mg/dL (9.00 mmol/L) and 275 mg/dL (7.11 mmol/L), respectively. These frequencies are in line with data from other study on Polish population, where this CNV change was found to be the most common. Chmara et al. also report that major *LDLR* rearrangements, as well as two-point variants in *LDLR* and *APOB* genes, are frequent causes of FH in Poland [20]; as such, CNVs are an important component of the genetic background of FH and should not be overlooked in the diagnostic process. Few studies have addressed CNV analyses in Polish populations, which may indicate a weak diagnostics rate in this area. The entire study group was characterized by very high levels of TC and LDL-c with mean concentrations of 388 mg/dL (10.03 mmol/L) and 318 mg/dL (8.22 mmol/L), respectively. These findings correlate with previous reports that heterozygous *LDLR* CNVs are associated with a more severe biochemical phenotype than other types of alteration [21].

The integration of the two methods proved to be particularly valuable in patient number 9, a five-year-old girl who presented a pathogenic missense variant and a two-exon deletion in *LDLR* gene, corresponding to a compound heterozygote state. Homozygous FH is a rare and life-threatening disease originally characterized by plasma cholesterol levels >500 mg/dL (>12.93 mmol/L), extensive xanthomas, and marked premature and progressive atherosclerotic cardiovascular disease (ACVD) [22]. The co-occurrence of high levels of TC and LDL-c with xanthomas in the patient corresponded to a homozygote state; this was suspected before the genetic testing and highly indicated a diagnosis of FH.

The missense c.1747C>T variant leads to histidine to tyrosine substitution at amino acid position 583, (p.His583Tyr). The variant has “conflicting interpretations of pathogenicity” with a significant preponderance of “pathogenic” ratings in ClinVar database. Combined computational analysis based on 10 predictor tools (BayesDel_addAF, DANN, DEOGEN2, EIGEN, FATHMM-MKL, LIST-S2, M-CAP, MVP, MutationTaster and SIFT) on Varsome (https://varsome.com, accessed on 6 June 2022) classified the above variant as pathogenic. This variant was previously reported to be one of the most common in FH patients in Singapore, Hong Kong, Taiwan and China [23]. In vitro research data, based on radioactive method, indicated that LDL receptor activity decreased by 60% due to c.1747C>T variant [23]. Based on the available reports, the detected variant should be considered as potentially pathogenic, disrupting the normal function of the LDL receptor.

Various single or multi-exon deletions in the *LDLR* gene have been reported so far (exons 2–3, exon 5, exon 7, exon 9–14) [21]; however, this is the first report of a heterozygous deletion of exons 9 and 10 co-occurring with SNV. Exons 7–14 of the *LDLR* gene encode the EGF-precursor homology domain, that plays a pivotal role in lipoprotein release during receptor recycling. The amino acid sequence in this domain is highly conserved, with 70–85% sequence homology shared across different species: human, cow, rabbit, hamster, rat, and the toad Xenopus laevis [21]. Any change within the domain may result in impaired release of LDL from the *LDLR* and potentially prevent the return of the receptor to the hepatocyte surface. The heterozygous deletion of two exons (9 and 10) likely resulted in no *LDLR* production (null allele) thus contributing to a massive reduction in *LDLR*-mediated endocytosis of LDL protein. The presented case is an interesting example of phenotypic-genotypic correlation in severe FH, illustrating the heterogeneity of the disease, and hence the need for comprehensive genetic diagnostics and early detection.

The MLPA technique is considered as the gold standard for CNV detection but for FH diagnostics, it is only well suited for *LDLR* gene analysis. The emerging reports of the presence of CNVs in *inter alia* the *PCSK9* gene [7], prompts the analysis of other FH-related genes not covered by commercially available MLPA kits. It is also a rather time-consuming and expensive method, and as such could not be applied for all patients in the present study. In addition, being a semiquantitative method, it also requires well-set standards and controls for correct peak analysis. 

It has always been a goal to find a unique method to detect both SNVs and CNVs from a single source of data. Many tools for CNV detection from NGS data have been developed but most show poor performance when dealing with small CNVs and were designed to work with whole-genome or whole-exome data [12]. Four different types of approach are currently used for detecting CNVs from NGS data: paired-end mapping-based detection (PE), split read based detection (SR), de novo assembly based detection (DA) and read depth based detection (RD) [24]. A large number of synthesized overlapping fragments is required to maximize DOC in panel NGS; this offers an opportunity to determine qualitative but also quantitative aspects of the selected genomic regions. This additional information from NGS data can be quantified using RD approaches, thus allowing for CNV detection [9]. Using the DECoN program, which is based exactly on RD, we were able to detect CNV changes in two new patients who had not been previously checked by the MLPA technique. Filtering all other patients, no changes were found in genes other than *LDLR*; nevertheless, our subsequent analysis of the data after panel NGS proved to be a useful tool. The increasing availability of clinical NGS panels and growing affordability of bioinformatic applications present a cost-effective opportunity to simultaneously identify the in-depth backgrounds of dyslipidemias.

## 5. Conclusions

To better understand the genetic background of inherited disorders, it is necessary to obtain increasingly advanced diagnostic tools to analyze the human genome. Our findings indicate that the compound genetic background of FH requires the implementation of several diagnostic tools, or preferably a single one that covers all possible types of genetic changes. Our results suggest that the NGS-based approach has the potential to identify large-scale variants in *LDLR* and could be further applied to extend CNV screening to other FH-related genes. Nevertheless, the outcomes from the bioinformatic approach still need to be confirmed by the MLPA method, and this remains the reference method for assessing CNV.

## Figures and Tables

**Figure 1 genes-13-01424-f001:**
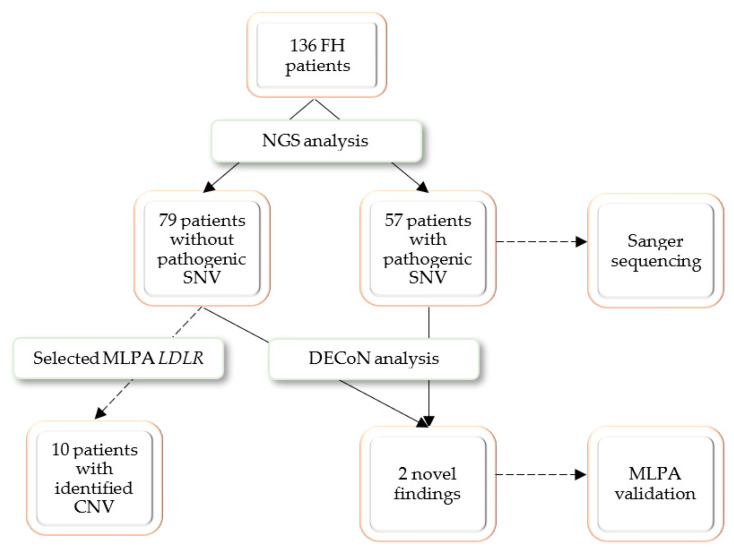
The schematic presentation of the conducted research.

**Figure 2 genes-13-01424-f002:**
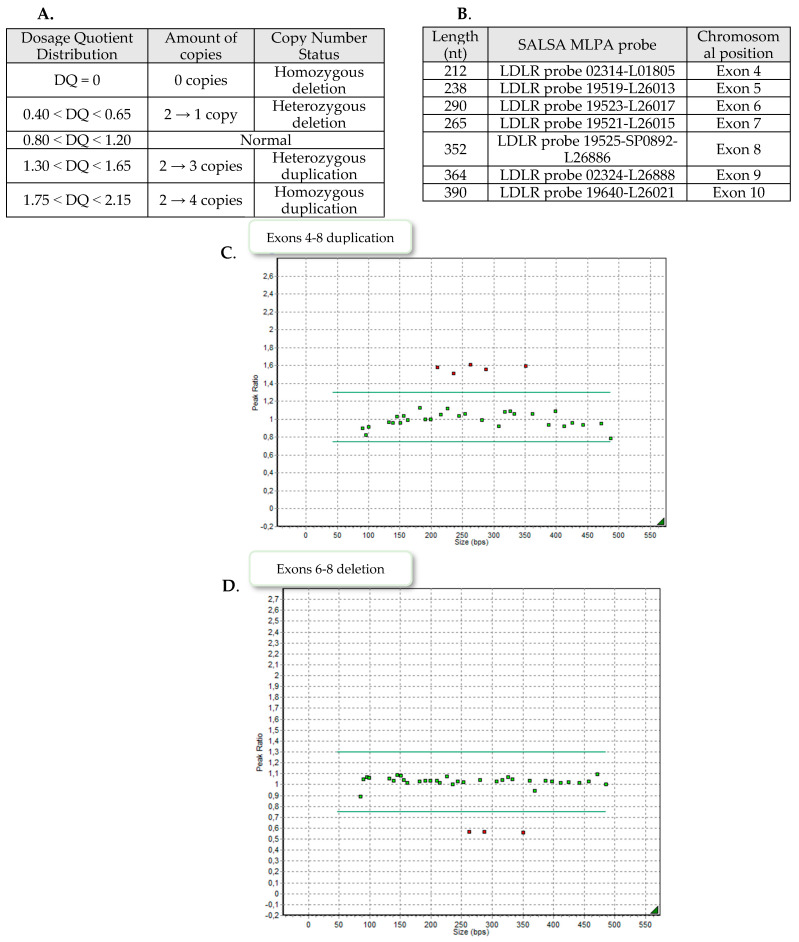
(**A**). Cut-off values for dosage quotient distribution, the equivalent number of copies and homo/heterozygosity state. (**B**). MLPA (multiplex ligation-dependent amplification) fragment length, SALSA MLPA probe numbers and their chromosomal position. (**C**). MLPA plots presenting exons 4–8 duplication of *LDLR*. The peak ratios of five probes oscillate in the range of 1.5–1.6 which corresponds to heterozygous duplication. (**D**). MLPA plots presenting exons 6–8 deletion of *LDLR*. The peak ratios of three probes oscillate in the range of 0.5–0.6 which corresponds to heterozygous deletion. (**E**). MLPA plots presenting exons 9–10 deletion of *LDLR*. The peak ratios of two probes oscillate in the range of 0.5–0.6 which corresponds to heterozygous deletion.

**Figure 3 genes-13-01424-f003:**
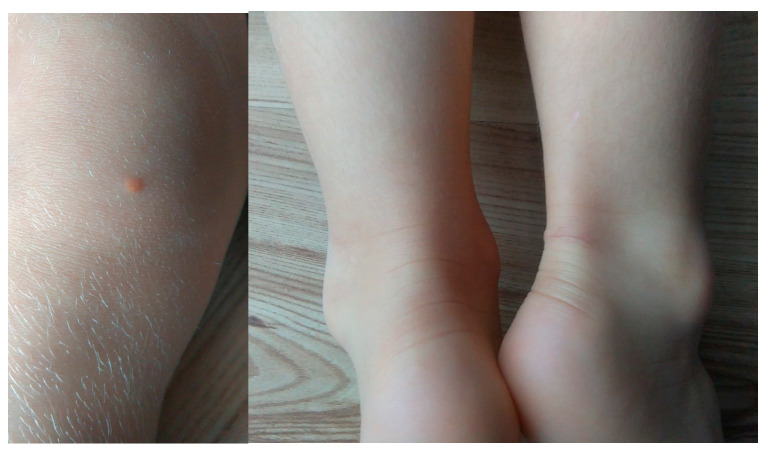
Xanthoma found on right knee. A similar-size xanthoma on the right Achilles tendon was removed.

**Table 1 genes-13-01424-t001:** Patients with identified FH-related CNV (copy number variation) and obtained genetic results.

No.	Sex	Age	TC [mg/dL]	LDL [mg/dL]	HDL [mg/dL]	TG [mg/dL]	NGS Result	MLPA Result
1.	Male	15	293	253	42	45	negative	Duplication of exons 4–8 of *LDLR*
2.	Female	50	287	203	41	NA	negative	Duplication of exons 4–8 of *LDLR*
3.	Female	3	277	222	47	43	negative	Duplication of exons 4–8 of *LDLR*
4.	Female	28	380	265	NA	NA	negative	Duplication of exons 4–8 of *LDLR*
5.	Female	64	424	344	32	344	negative	Duplication of exons 4–8 of *LDLR*
6.	Female	14	336	263	52	106	negative	Deletion of exons 6–8 of *LDLR*
7.	Female	35	368	282	59	134	negative	Deletion of exons 6–8 of *LDLR*
8.	Female	16	440	367	43	144	negative	Duplication of exons 4–8 of *LDLR*
9.	Female	5	745	693	NA	NA	positive	Deletion of exons 9–10 of *LDLR*
10.	Female	68	* 196	* 140	* 37	* 104	negative	Duplication of exons 4–8 of *LDLR*
11.	Male	28	333	290	39	93	negative	Duplication of exons 4–8 of *LDLR*
12.	Female	10	371	274	83	70	negative	Duplication of exons 4–8 of *LDLR*

* Lipid parameters obtained during the hypolipidemic treatment. The patient was excluded from a statistical summaries.

**Table 2 genes-13-01424-t002:** Results obtained with the DECoN program. The following columns indicate: Correlation—the maximum correlation between the test sample and any other sample in the full set of BAM files, N.comp—the number of samples used as the reference set, Start.b—the number of the first exon in the call from the analyzed BED file, End.b—the number of the last exon in the call from the analyzed BED file, CNV type—type of copy number variation change, Genomic ID—genomic coordinates of detected variant according to GRCh38, BF—the Bayes factor associated with the call, Reads.expected—the number of expected reads under the probabilistic model, Reads.observed—the number of observed reads, Reads.ratio—the ratio of observed to expected reads and Gene name.

No.	Correlation	N.Comp	Start.b	End.b	CNV Type	Genomic ID	BF	Reads. Expected	Reads. Observed	Reads. Ratio	Gene
11.	0.997068559377578	2	250193	250214	duplication	chr19:11105220-11111639	58.5	8354	10869	1.3	*LDLR*
12.	0.998912665575895	12	250193	250214	duplication	chr19:11105220-11111639	115	5100	7114	1.39	*LDLR*

## Data Availability

The data presented in this study are openly available in ClinVar (www.ncbi.nlm.nih.gov/clinvar/, accessed on 6 June 2022), DECIPHER (www.deciphergenomics.org, accessed on 6 June 2022) and Varsome (https://varsome.com, accessed on 6 June 2022) database.

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
