# Peer review of "Identification of New Copy Number Variation and the Evaluation of a CNV Detection Tool for NGS Panel Data in Polish Familial Hypercholesterolemia Patients"

_genes, 2022, doi:10.3390/genes13081424_

Round 1

Reviewer 1 Report

The authors aimed of to identify novel CNV changes in FH patients without detected pathogenic SNVs using targeted sequencing and evaluation of CNV calling tool (DECoN) working on gene panel NGS data in a group of 136 adult and child patients with possible FH. They also assessed its suitability as a screening step in genetic diagnostics. To test the DECoN program, the whole study group was referred for bioinformatic analysis. The DECoN program detected duplication of exons 4-8 in the LDLR gene in two patients, whose genetic analysis was stopped after the NGS step. The integration of the two methods proved to be particularly valuable in a five-year-old girl presenting with extreme hypercholesterolemia, with both a pathogenic missense variant (c.1747C>T) and exons 9-10 deletion in LDLR. This is the first report of a heterozygous deletion of exons 9 and 10 co-occurring with SNV. They concluded that the NGS-based approach has the potential to identify large-scale variation in the LDLR gene and could be further applied to extend CNV screening in other FH-related genes. Nevertheless, the outcomes from the bioinformatic approach still need to be confirmed by MLPA; hence, the latter remains the reference method for assessing CNV in FH patients.

It is a well designed and nicely presented study presenting very important novel data on screening of CNV changes in FH patients.

Comments:

·       1. Some clinical consequences, especially therapeutic difficulties in these severe cases could be mentioned.

·       2. The relatively low number of patients with CNV changes is obvious, it should be mentioned as a limitation of the study.

·       3. The figure legend of Fig 1, especially Fig1C-E should be completed.  

·       4.   The reference style is not appropriate.

Reviewer 2 Report

congratulations for the this scientific effort to address the essential part of resrch

Methodology can be improved not necessarily 
